# Hollow Porous CoO@Reduced Graphene Oxide Self-Supporting Flexible Membrane for High Performance Lithium-Ion Storage

**DOI:** 10.3390/nano13131986

**Published:** 2023-06-30

**Authors:** Junxuan Zhang, Jie You, Qing Wei, Jeong-In Han, Zhiming Liu

**Affiliations:** 1Flexible Display and Printed Electronics Laboratory, Department of Chemical and Biochemical Engineering, Dongguk University-Seoul, Seoul 04620, Republic of Korea; zjx950518@gmail.com; 2Shandong Engineering Laboratory for Preparation and Application of High-Performance Carbon-Materials, College of Electromechanical Engineering, Qingdao University of Science & Technology, Qingdao 266061, China; yj1048154827@163.com (J.Y.); qing3303@163.com (Q.W.); 3Qingdao Institute of Bioenergy and Bioprocess Technology, Qingdao Industrial Energy Storage Research Institute, Chinese Academy of Sciences, Qingdao 266101, China

**Keywords:** CoO@rGO, graphene, MOF, flexible electrodes, hollow structure, ultrafast integration, lithium-ion batteries

## Abstract

We report an environment-friendly preparation method of rGO-based flexible self-supporting membrane electrodes, combining Co-MOF with graphene oxide and quickly preparing a hollow CoO@rGO flexible self-supporting membrane composite with a porous structure. This unique hollow porous structure can shorten the ion transport path and provide more active sites for lithium ions. The high conductivity of reduced graphene oxide further facilitates the rapid charge transfer and provides sufficient buffer space for the hollow Co-MOF nanocubes during the charging process. We evaluated its electrochemical performance in a coin cell, which showed good rate capability and cycling stability. The CoO@rGO flexible electrode maintains a high specific capacity of 1103 mAh g^−1^ after 600 cycles at 1.0 A g^−1^. The high capacity of prepared material is attributed to the synergistic effect of the hollow porous structure and the 3D reduced graphene oxide network. This would be considered a promising new strategy for synthesizing hollow porous-structured rGO-based self-supported flexible electrodes.

## 1. Introduction

Due to the increasing demand for higher capacity, higher power density, longer cycle life of energy storage devices for portable electronic devices, wearable electronic devices, and energy-consuming devices such as electric vehicles, research on new lithium-ion batteries with higher performance and more excellent safety has become more and more urgent [1,2,3,4]. Lithium-ion batteries have significant advantages and have become the first choice for the modern day owing to their high energy density, high cycle life, and high efficiency [5,6,7]. The low theoretical capacity of conventional graphite anode materials contradicts the need for higher-capacity Li-ion batteries [8,9,10].

Transition metal oxides (TMOs), the most promising active anode materials for the new generation of Li-ion batteries, have received much attention from researchers due to their high specific capacity and exemplary safety [11,12]. Among them, CoO has been further investigated due to its high theoretical capacity (716 mAh g^−1^), relatively low cost, and fully reversible electrochemical reaction. However, transition metal oxides, including CoO, are conversion-type anode materials that generate large volume expansion during cycling, leading to the chalking of active material particles and greatly reduced cycle life. Thus, the practical application of metal compounds for lithium-ion batteries is severely limited [13,14,15,16]. Building nanostructures is an effective solution to improve their performance [17,18]. Metal–organic backbone (MOF) derivatives can effectively maintain the morphology of MOF precursors due to their tunability at the molecular level and unique porous backbone structure [19,20]. Therefore, MOF is widely used as a template [21,22], and TMO nanostructures such as CoO can be fabricated by a simple process [23]. However, the low electrical conductivity and slow reaction kinetics of CoO nanomaterials still limit their performance in lithium-ion battery electrode materials.

Graphene is often used as an ideal substrate material due to its high electrical conductivity, huge specific surface area, and excellent physical and mechanical flexibility [24,25,26]. The composite of graphene with TMOs is considered a reasonable and effective solution at present. On the other hand, it can also effectively alleviate the chalking of the active material caused by the volume expansion during cycling, thus greatly improving the battery cycling stability [27]. To date, various composites of TMOs with graphene, including Fe_3_O_4_ nanoflakes/RGO composites [28], MnO_2_/reduced Graphene Oxide Nanosheet [29], hollow Fe_3_O_4_/graphene hybrid films [30], and Co_3_O_4_@rGO nanocomposites [31], etc., have been successfully reported for lithium-ion battery electrodes. However, these reported TMOs/graphene composites usually have more preparation steps, high preparation cost, and complicated processing, which seriously hinder their industrial application in Li-ion batteries.

Herein, we designed and synthesized a simple, environmentally friendly, cost-effective CoO@rGO flexible membrane electrode. Co-MOF is known as a metal–organic skeleton and its derivatives can effectively maintain its intrinsic morphology, and its composite with a three-dimensional (3D) graphene network has also exhibited excellent electrochemical properties. In addition, it has been shown that ammonia has an etching effect on some MOFs, which can be etched into hollow materials [28], and at the same time, graphene oxide is rapidly reduced. The hollow material is more favorable for lithium ion transport and storage, so the composite of hollow structured CoO with graphene oxide would be an excellent self-supporting anode material.

## 2. Material and Methods

### 2.1. Synthesis of Co-MOF

In the typical synthesis of Co-MOF, 0.6 mmol cobalt acetate tetrahydrate and 0.9 mmol sodium citrate were dissolved in 20 mL of deionized (DI) water to obtain solution A. In addition, 0.4 mmol potassium cobalt cyanide was dissolved in 20 mL of deionized water to obtain solution B. The solution A and solution B were then quickly mixed together and stirred for 12 h [32]. The precipitated product was collected by centrifugation and washed three times with deionized water and alcohol, followed by vacuum drying overnight at 70 °C to obtain the Co-MOF precursor material.

### 2.2. Synthesis of Co-MOF@rGO and CoO@rGO Flexible Membranes

A total of 20 mg of GO and 60 mg of Co-MOF precursor were dissolved into 5 mL DI water. The precursor solution was mixed well by stirring for 10 min, poured into disposable Petri dishes, and then freeze-dried to obtain Co-MOF@GO membrane material. Then, hot (NH_4_)_2_S solution was added dropwise to the above membrane for rapid reduction. After 1 min, the excess (NH_4_)_2_S was quickly washed off with deionized water and freeze-dried to obtain Co-MOF@rGO flexible membrane finally.

The above-obtained Co-MOF@rGO flexible film was heat treated at 550 °C for 2 h under Ar atmosphere with a heating rate of 2 °C min^−1^, and the CoO@rGO flexible film could be directly used as an electrode after natural cooling.

### 2.3. Material Characterization

The morphology of the products was characterized by field emission scanning electron microscopy (SU8010, HITACHI, Japan), transmission electron microscopy (TEM), and high-resolution TEM (HRTEM). Energy dispersive X-ray spectroscopy (EDX) analysis and corresponding elemental mapping were performed using an X-ray spectrometer attached to the TEM instrument. The crystalline phases of the products were collected by X-ray diffraction (MiniFlex600, Rigaku, Japan) using monochromatic Cu-Kα lines as a radiation source. The thermal stability of the samples in the air was evaluated by thermogravimetric analysis (209-F3, NETZSCH, Germany) in the temperature range of 40–800 °C, with a ramp rate of 10 °C min^−1^. Raman spectra were obtained by testing on a Raman spectrometer (1PQG99, Renishaw, Korea) at a laser wavelength of 532 nm.

### 2.4. Electrochemical Measurements

CoO@rGO and rGO flexible films were cut into small 10 mm diameter discs and used directly for an electrochemical evaluation. As a comparison, CoO electrodes were prepared by mixing 80 wt% of active electrode material (CoO), 10 wt% of carbon black, and 10 wt% of polyvinylidene fluoride binder to prepare a slurry. Then, the slurry was uniformly coated on the copper foil and dried overnight in a vacuum oven at 60 °C. The CR2032 half-cells were assembled in an argon-filled glove box by cutting into 10 mm-diameter disc electrodes by a button cell slicer, using lithium sheets as counter electrodes and a mixture of vinyl carbonate (EC) and diethyl carbonate (DEC) with 5% fluoroethylene carbonate (FEC) as the electrolyte. Cycling performance and rate performance tests were performed on a cell test system (LANHE, CT2001A, Wuhan, China) with a voltage range of 0.01–3.0 V. Cyclic voltammetry tests with a voltage window of 0.01–3.0 V (vs. Li^+^/Li, 0.1 mV s^−1^) were performed on an electrochemical workstation (CHI, 760E, Shanghai, China).

## 3. Results and Discussion

### 3.1. Characterization

As shown in Figure 1, the Co-MOF nanocubes (Appendix A) prepared in advance were co-dispersed with GO in deionized water and later freeze-dried to form a film directly. Then, using the strong reduction effect of (NH_4_)_2_S, GO was rapidly reduced to rGO, and the Co-MOF@rGO flexible film was obtained. It is noteworthy that Co-MOF is a tunable and unique porous backbone structure at the molecular level, and in addition, it can be etched by NH_4_^+^ to form a hollow structure. This is because ammonia will coordinate with cobalt cations to form complex ions, which cause the etching of cobalt [33]. Therefore, while the three-dimensional reduced graphene oxide network improves the electrical conductivity of the material, its hollow structure further shortens the ion transport path; thereby, the ion transport efficiency can be effectively improved. Meanwhile, the buffering effect of the 3D reduced graphene oxide network can effectively adapt to the volume change during the cycling process. Thus, it will be a promising new anode material for energy storage devices.

The flexible membrane electrodes prepared by this promising synthetic strategy are shown in Figure 1a and Appendix A, which can well meet the requirements of bendability and wearability. Appendix A shows the SEM images of the prepared Co-MOF nanocubes, and all the cubes show smooth surface structures with a size of about 1 μm. The SEM images of the Co-MOF@GO composites were obtained after compounding with graphene oxide and are shown in Figure 1b,c. The Co-MOF nanocubes are uniformly wrapped in a 3D network of graphene oxide, showing a good 3D buffer structure. In addition, it can be seen that the Co-MOF@GO composite keeps its initial morphology intact and is successfully compounded with graphene oxide before being etched by ammonium sulfide.

When ammonium sulfide is added, GO is rapidly reduced to rGO, as shown in Figure 1d,e, and the original oxygen-containing functional groups in graphene oxide are destroyed, thus contracting and cross-linking each other and tightly wrapping the Co-MOF nanocubes. It is noteworthy that the Co-MOF in the obtained Co-MOF@rGO flexible composite film produced a slight deformation due to the NH_4_^+^ in ammonium sulfide, which simultaneously had a strong etching effect on the Co-MOF nanocubes; it was etched into a hollow structure in a very short period of time. As can be seen from the SEM images at high magnification in Figure 1e, the etched Co-MOF nanocubes exhibit a different contrast around the etched area from the central position, indicating that they may have been etched into hollow structures. This judgment is confirmed by the subsequent SEM of individual broken Co-MOF nanocubes, as shown in Figure 1f,g. It can be seen from the broken Co-MOF nanocubes that the Co-MOF nanocubes are hollow inside, leaving a layer of outer wall. This proves the strong etching effect of ammonium sulfide on Co-MOF nanocubes and illustrates the successful synthesis of hollow Co-MOF nanocubes with rGO-based flexible composites.

The Co-MOF@rGO flexible composite film was heat-treated, and the Co-MOF was heat-treated to obtain its corresponding derivatives, which were then used as electrode materials. The SEM images of the obtained CoO@rGO flexible composite films are shown in Figure 1h,i and Appendix A, from which it can be seen that the CoO nanocubes derived from the Co-MOF nanocubes obtained after heat treatment still maintain the complete cubic structure with a size of about 1 μm. Notably, the CoO nanocubes obtained after heat treatment have a hazy, gauzy texture and exhibit a more sparse and porous surface structure. This may be attributed to the porous structure caused by the escape of gases generated during the heat treatment process. This porous structure can further reduce the ion transport path and promote rapid lithium-ion transport, which is conducive to better lithium-ion insertion and extraction, and synergize with the high electrical conductivity of the 3D reduced graphene oxide network and the excellent buffer structure, thus effectively improving the cycling stability while also effectively improving the composite multiplicity performance. Appendix A shows the EDS analysis of the CoO@rGO flexible composite film. As expected, the Co and O elements in the obtained CoO@rGO flexible composite film are uniformly distributed and correspond to the CoO nanocubes. At the same time, the C elements corresponding to the reduced graphene oxide around the CoO nanocubes were also uniformly distributed around the CoO nanocubes, and they wrapped the CoO nanocubes tightly. In addition, the S and N heteroatom doping introduced by adding ammonium sulfide is also present in the heat-treated CoO nanocubes, further demonstrating the successful composite of CoO@rGO flexible composite film materials.

The internal structure of Co-MOF nanocubes in the Co-MOF@rGO flexible composite film was further investigated by TEM analysis, as shown in Figure 2a,b. It is obvious that the Co-MOF nanocubes etched by ammonium sulfide exhibit a distinct hollow structure with an empty internal structure and an outer wall thickness of about 155 nm (Figure 2c). Appendix A shows the tight junctions between the rGO and CoO crystals, and the 0.24 nm lattice stripe labeled in the figure is attributed to the CoO(111) crystal plane. And the elemental distribution in the Co-MOF@rGO flexible composite film was analyzed in Figure 2d, from which the hollow structure of the hollow Co-MOF nanocubes could be seen, (e), (f), (g), (h), (i) are the elemental mapping of C, Co, O, S and N, respectively.

The elemental distribution shows that Co and O elements are uniformly distributed on the outer wall of the hollow Co-MOF nanocubes, and C elements can also be seen around the hollow Co-MOF nanocubes due to the peripheral wrapping of the reduced graphene oxide. In addition, the introduction of S and N elemental heteroatoms can also be seen on the elemental distribution map due to the reduction and etching of ammonium sulfide, which is beneficial for the storage of lithium ions. From this elemental mapping analysis, it can be well illustrated that the Co-MOF@rGO flexible composite film material is successfully compounded.

To further determine the composition of the hollow porous CoO@rGO flexible composite films, the resulting composites were characterized by XRD. As shown in Figure 3a, all major diffraction peaks of the hollow porous CoO@rGO flexible composite film can be successfully indexed to the corresponding standard card (JCPDS 48-1719). The hollow porous CoO@rGO flexible composite membrane shows very distinct diffraction peaks at 36.74, 42.61, 61.62, 73.97, and 77.65°, corresponding to the (111), (200), (220), (311), and (222) crystal planes of cobalt oxide, respectively. The broad diffraction peaks centered at 2θ ≈ 26° in the diffraction pattern of the hollow porous CoO@rGO flexible composite film is be attributed to the typical (002) crystal plane of rGO [34], and the absence of other impurity peaks is a good proof of the successful synthesis of the hollow porous CoO@rGO flexible composite film.

In order to obtain the carbon content in the hollow porous CoO@rGO flexible composite film, thermogravimetric analysis was also performed and represented in Figure 3b, and the mass content of rGO in the hollow porous CoO@rGO flexible composite film can be calculated from the thermogravimetric analysis results. During the heating process from 40–800 °C under air atmosphere, carbon is gradually decomposed to form carbon dioxide and expelled, while CoO nanocubes are gradually oxidized to Co_3_O_4_ when heated in the air [35]. It can be seen that at 800 °C, 49.08 wt% content of Co_3_O_4_ is finally obtained, so it can be calculated that the hollow porous CoO@rGO flexible composite film has a mass content of CoO of about 45.82%, which leads to the further calculation that the mass content of rGO is about 54.18%.

As shown in Figure 3c, the Raman spectra of CoO@rGO 3D mesh composites have two main peaks at 1349 cm^−1^ and 1574 cm^−1^ for the D-band of the A_1g_ vibrational mode used for disordered carbon and the G-band of the E_2g_ vibrational mode for ordered graphitic carbon. The D to G band’s intensity ratio (I_D_/I_G_) usually reflects the degree of defects and disorder in carbon materials [36]. The intensity ratio (I_D_/I_G_) of the D-band to the G-band of the CoO@rGO composite is about 1.37, indicating that there are many defects in the CoO@rGO 3D mesh composite. Compared with the Raman spectrum of the rGO flexible film (Appendix A), the CoO@rGO flexible composite film has a stronger D-band intensity, indicating more defects or disorder sites in the heterogeneous structure, which helps to provide more active regions for lithium storage. Figure 3d shows the FTIR spectra of CoO@rGO, which exhibits characteristic stretching frequencies at 823, 1039, 1385, 1762, and 3343 cm^−1^, corresponding to C=C, Co-O-Co, S-O, C=O, and O-H functional groups, respectively. One of the peaks at 1385 cm^−1^ represents the S-O stretching vibration of the sulfate group, which can be attributed to the addition of ammonium sulfide etching in the Co-MOF@GO composite. The above analysis confirms the successful preparation of CoO@rGO 3D mesh composites.

As shown in Figure 4a,b, the adsorption–desorption experiments of N2 were performed on co-MOF@rGO before and after etching, and the pore size distribution was calculated by the BJH method. The type IV isotherm plot with a distinct characteristic hysteresis loop after etching (Figure 4b) indicates the dominant number of mesopores compared to the unetched sample. The obtained specific surface area is 22.4 m^2^ g^−1^, and the pore size distribution is in the range of 2–20 nm. It is noteworthy that the specific surface area of the etched composites increases compared to the pre-etching composites possessing a specific surface area of 11.7 m^2^ g^−1^. This is due to the coupling effect between the N,S doping introduced during the etching process and the hollow structure produced by the etching. The large specific surface area helps to improve the utilization of the active electrode material, while the abundant mesopores of the material facilitate the rapid diffusion of lithium ions and increase the lithium storage sites. X-ray photoemission spectroscopy (XPS) measurements show the surface chemistry of CoO@rGO flexible films. The XPS spectra in Figure 4c confirm the presence of C, CO, O, N, and S elements in the CoO@rGO flexible film. The low content of N and S elements can be attributed to the introduction of heteroatoms caused by the reduction of (NH_4_)_2_S during the etching process. The introduction of N and S into the carbon structure can add more active sites, thus increasing the lithium storage capacity and providing high specific capacity and rate performance for Li-ion batteries. As shown in the C 1s spectrum (Figure 4d), the XPS C 1s spectrum of CoO@rGO flexible film can be divided into two main peaks, where 284.76 eV corresponds to the C=C/C–C bond and 286.32 eV corresponds to the C–O bond. The peak of C 1s is attributed to rGO. The stronger peak with a binding energy of 284.76 indicates the deoxygenation process accompanying rGO reduction in CoO@rGO flexible films, which is consistent with the previously reported results [32]. In the fitted high-resolution Co 2p spectra (Figure 4e), the two peaks located at 781.26 and 797.05 eV can be attributed to Co^2+^ with Co 2p_3/2_ and Co2p_1/2_, which indicates the presence of Co^2+^ in the CoO@rGO flexible film, consistent with its theoretical chemical state. The high-resolution O1spectrum (Figure 4f) can be decomposed into three peaks corresponding to Co–OH (533.68 eV), Co–O–C (531.83 eV), and Co–O (530.29 eV), respectively. One of them at the binding energy of 531.83 eV corresponds to the Co–O–C bond of the oxygen-containing functional group on the surface of rGO. This result indicates that CoO and rGO successfully hybridize through Co–O–C, which provides a strong binding strength at the mutual interface of CoO and rGO [37].

### 3.2. Electrochemical Performances

Our designed CoO@rGO flexible film has an excellent three-dimensional network that facilitates overcoming significant volume variations, promotes rapid lithium-ion transport, and provides reasonable rate capability and cycling performance for the assembled Li-ion batteries. More importantly, benefiting from its flexible self-supporting feature, the CoO@rGO flexible film can be quickly prepared and directly utilized as a stand-alone negative electrode for Li-ion batteries and avoids using conductive binders. We have assembled CoO@rGO flexible films into half cells and evaluated their electrochemical performance. Figure 5a shows the charge and discharge curves at a current density of 1 A g^−1^ with a voltage window of 0.01–3.0 V. It is clear that the first turn discharge and charge capacities of the CoO@rGO flexible membrane electrode are about 1020 mA h g^−1^ and 722 mA h g^−1^, respectively, thus corresponding to an initial coulombic efficiency (ICE) of 72%. The irreversible capacity loss can be attributed to the formation of the SEI membrane, some undecomposed Li_2_O, and the irreversible decomposition of the electrolyte [38]. The rate performance of the CoO@rGO flexible-film electrode was evaluated by gradually increasing the current density from 0.1 to 2.0 A g^−1^ (Figure 5b). It can be seen that the CoO@rGO flexible film exhibits the best rate of electrochemical performance compared to the comparison sample. Reversible capacities of 556, 437, 373, 336, 320, 307, and 251 mA h g^−1^ were obtained at 0.1, 0.2, 0.4, 0.6, 0.8, 1.0, and 2.0 A g^−1^, respectively. Moreover, the specific capacity can be easily recovered to 509 mA h g^−1^ when the current density is reduced to 0.1 A g^−1^, proving its excellent rate performance. In contrast, the comparison electrodes CoO and pure rGO films exhibit relatively low specific capacities and do not recover their initial specific capacities well when the current density is restored to the initial current.

To explain the excellent kinetic origin of CoO@rGO, CV measurements at different scan rates were used to evaluate the kinetic reactions (Figure 5c). In the CV curve, the scan voltage from negative to positive can be seen as the anodic oxidation process, which corresponds to the oxidation peak, and vice versa as the cathodic reduction process, which corresponds to the reduction peak. It can be seen that when the scan rate is at 0.1 mVs^−1^, the cathodic and anodic peaks in the CV curves are symmetrical in shape, and the peak heights are basically the same, indicating that the electrochemical reactions occurring at the CoO@rGO flexible-film electrode/electrolyte interface have good reversibility. With the increase in scanning rate, all CV curves exhibit similar shapes and well-preserved redox peaks, with peak current (i) increasing with scan rate (v). The relationship between peak current and scan rate is i = av^b^, where a and b are parameters, and b is worth calculating by plotting a graph and fitting a straight line [39,40,41]. Determining the value of b allows a qualitative analysis of the charge storage mechanism. When the value of slope (b) is close to 0.5, the storage behavior is dominated by diffusion-type reactions; when it is close to 1.0, it tends to capacitively controlled processes [42]. The b-values of the anodic and cathodic peaks of the CoO@rGO flexible film are 0.84 and 0.82 (Figure 5d), suggesting that it is dominated by capacitance-controlled behavior. Specifically, the ratio between diffusion-controlled and capacitive-controlled can be further quantified with a relationship given by the Equation [43,44,45]:i = k_1_ ν + k_2_ ν^0.5^. 

With this equation, the ratio of pseudocapacitive charge storage can be obtained for different scan rates. Figure 5e shows the capacitive contribution (68%) to the entire scan area at 0.2 mV s^−1^. As shown in Figure 5f, when the CoO@rGO flexible film was tested at scan rates of 0.2, 0.4, 0.6, 0.8, and 1 mV s^−1^, the contribution of capacitive behavior was 68%, 72%, 77%, 83%, and 93%, respectively. It can be seen that the contribution of capacitance increases with increasing scan rate. The high pseudo-capacitance rate may be due to the synergistic effect of the hollow structure formed by Co-MOF after etching by ammonium sulfide and after compounding with reduced graphene oxide, which provides more active sites for Li intercalation/delamination as well as lithium storage.

Figure 5g shows the cycling performance of the CoO@rGO composite electrode compared to the comparison electrode CoO and pure rGO films at a current density of 1 A g^−1^. It can be seen that the CoO@rGO composite electrode exhibits the most excellent cycling performance compared to the comparison electrode. The CoO@rGO composite electrode shows a trend of consistently increasing capacity over one hundred and fifty cycles during the entire charge/discharge process with the voltage window of 0.01–3.0 V at 1 A g^−1^ current density, which is common for various nanostructured metal oxide electrodes whose capacity increases with cycling [46]. Activation of the electrode material during charge/discharge is one of the main reasons for this phenomenon, and the gradual increase in capacity may be due to the reversible growth of the polymer gel-like film caused by the degradation of the dynamically active electrolyte [47,48]. In addition, the hollow structure of CoO effectively restrains the volume expansion during cycling, while the rGO sheet layer provides an excellent buffering effect. After 600 cycles, the CoO@rGO composite electrode exhibits an impressive discharge capacity of 1103 mA h g^−1^. In contrast, the discharge capacities of CoO and pure rGO electrodes are only 483 and 155 mA h g^−1^ (Appendix A). This is significantly better than many other reported cobalt-based anode materials for lithium-ion batteries (Appendix A). In addition, from three cycles onwards, for the CoO@rGO composite electrode, around 97% coulombic efficiency remains stable, showing its excellent reversible Li^+^ insertion extraction performance. This fully reflects the important role of the hollow structure of the CoO and rGO three-dimensional conductive network for the excellent cycling performance of the CoO@rGO flexible film self-supported electrode.

## 4. Conclusions

In summary, we designed and synthesized a flexible self-supporting film electrode with hollow porous CoO nanocubes compounded with rGO. The CoO@rGO flexible-film composite electrode prepared using this method has many advantages: it can be prepared faster and more economically, on a large scale, with a simple process flow, which is beneficial for its industrial application in lithium-ion batteries, and the method is self-forming without adding conductive agents and binders, which can effectively improve the overall energy density of the electrode. The hollow and porous structure can effectively shorten the ion transport path and provide more active sites for lithium ions. In addition, the large-sized rGO sheet layer not only improves the overall conductivity of the material and promotes rapid charge transfer but also provides sufficient buffer space for the hollow Co-MOF nanocubes. Due to the synergistic effect of this hollow porous structure and the three-dimensional reduced graphene oxide network, the CoO@rGO flexible-film electrode shows extremely excellent electrochemical performance. We investigated the electrochemical performance of CoO@rGO flexible film as an anode material for Li-ion batteries, showing good cycling stability (1103 mA h g^−1^ after 600 cycles at 1.0 A g^−1^). Even at a high current density of 5.0 A g^−1^, a reversible capacity of 586 mA h g^−1^ was maintained, which is significantly better than many previously reported cobalt-based electrode materials. The CoO@rGO flexible-film composite electrode is expected to be an ideal next-generation anode candidate for industrial applications of lithium-ion batteries.

## Data Availability

The data presented in this study is available on request from the corresponding authors.

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
