# Peer review of "Hollow Porous CoO@Reduced Graphene Oxide Self-Supporting Flexible Membrane for High Performance Lithium-Ion Storage"

_nanomaterials, 2023, doi:10.3390/nano13131986_

Round 1
Reviewer 1 Report
I have reviewed the submitted article entitled “Hollow porous CoO@reduced graphene oxide self-supporting flexible membrane for high-performance lithium-ion storage” to Nanomaterials/MDPI by Zhiming Liu et al. Lithium-ion batteries are now dominating due to their excellent electrochemical performance; however, commercial graphite as anode greatly restricts their practical applications in large-scale energy storage devices. Therefore, it is urgently necessary to search for other high-performance anode materials to replace the graphite anode. The authors in the submitted work explored the concept of CoO@rGO as an alternative to graphite. The explored concept of using carbon nanostructures is not new it has been reported for Li-S batteries but could be new for Li-ion using Co-MOF as self-supporting anodes.
The area of research is significant and addresses the correlation of the self-supporting structure with the activity of electrochemical energy storage with the synergistic effect suitable for cycling stability in battery applications. The manuscript is written well with a well-organized story, but the main concern of this paper and the areas that require clarification are given below.
1. Does the Cobalt are well anchored on the rGO with strong interfacial interaction?
2. What would be the reversible capacity energy storage when the cell tested at a very high current density of say 5 A/g?
3. What is the advantage of hollow porous structures and also have a synergistic effect from different components such as self-supporting nanostructure, Co, rGo, etc.?
4. Does the presence of rGO effectively accommodate the volume change?
5. How does the obtained reversible capacity compare to other reported anode materials?
6. Page 2 lines 49 and the next paragraph starting in line 50 do not have a smooth flow in the story. There is a jump, please smooth it.
7. Section 2.1 What is solutions A and B refer to?
8. The pseudo capacitance (shown in Figure 4) and the formation of nanocubes (shown in Figure 1) reported for similar ternary metal oxides using CoO in the literature such as doi.org/10.1021/acsomega.9b03657; and doi.org/10.3390/nano7110356 must be included in the discussion.
9. Page 5, line 147, what is “muslin” granularity>
10. Why are the TEM images showing distorted cubes?
11. Page 8, the paragraph related to kinetics has been already well divulged by Kethaki Wickramaarachchi et al in 2022. Therefore, please articulate it appropriately referring to the article.
12. Figure 4a is better to remove the phrase “ICE 72” as it has been said in the text. Appears as though it is a sample name but not.
13. The shape of the CV curves in Figure 4c must be detailed.
14. Please make the conclusion qualitative by bringing the values.
Some minor corrections (as given in my comment) are required.
Reviewer 2 Report
The publication entitled, ‘Hollow porous CoO@reduced graphene oxide self-supporting flexible membrane for high performance lithium-ion storage’ by Zhang et al., describes a method towards the fabrication of hollow Co-MOF combined with GO for improved Li ion mobility. The work is interesting and the samples have been well characterized. Please consider the following concerns.
(1) The introduction is too generic. Please stick to works related to this present study to introduce your paper. Please be more specific and discuss other peoples works with regards to synthesis and electrochemical properties.
(2) The authors should explain why hollow cubes work better than filled cubes for ion transport. What is the mechanism that governs such transport in both structures? Appropriate references should be used.
(3) The letters in the EDX maps are not visible. Basically, this reviewer is unable to see which map corresponds to which element.
(4) It is not clearly explained in the manuscript. Please explain how only the inside of the CoO cube is etched with ammonia?
(5) What is the effect of adding ammonia to GO or CoO? They will introduce N doping. How does that effect the electrochemical properties?
(6) What is the actual size of the CoO-MOF? The EDX maps were carried out on ~1 micrometer sized cube. However, figure 2d shows a 10 nm nanoparticle surrounded by graphene and is not hollow. What is the reason for the discrepancy in size and morphology?
(7) On the XRD pattern, please index RGO and and CoO peaks.
(8) It is important to know whether after a heat treatment of the MOF Co2C is present. This is highly likely. It could also explain better transport properties as it has a metallic structure. FTIR can not provide this information. I recommend that the authors perform XPS.
(9) Please explain why the electrochemical performance is better in hollow structures. Li ion adsorption desorption is a surface phenomenon. Can the authors perform BET to analyze the specific surface before and after ammonia etching?
(10)Please tabulate parameters obtained from electrochemical performances. It is very difficult to follow the different adsorption and diffusion related mechanisms by reading alone.
Some sentences are not properly formulated. These errors can be easily corrected. Please ask someone proficient to read the paper.
Round 2
Reviewer 1 Report
The revised version is suitable for publication.
Reviewer 2 Report
The authors have performed all the revisions requested by this author. I asked if they were able to do BET but were not obliged to. Nevertheless, they have done it and it brings added value to the paper. The paper can be published.
very minor typos.